# RED: EFFICIENTLY BOOSTING ENSEMBLE ROBUSTNESS VIA RANDOM SAMPLING INFERENCE

## ABSTRACT

Despite the remarkable achievements of Deep Neural Networks (DNNs) in handling diverse tasks, these high-performing models remain susceptible to adversarial attacks. Considerable research has focused on bolstering the robustness of individual models and subsequently employing a simple ensemble defense strategy. However, existing ensemble techniques tend to increase the inference latency and the parameter number while achieving suboptimal robustness, which motivates us to reconsider the framework of model ensemble. To address the challenge of suboptimal robustness and inference latency, we introduce a novel ensemble defense approach called Random Ensemble Defense (RED). Specifically, we expedite inference via random sampling, which also makes it difficult for an attacker to attack a model ensemble. To effectively train a model ensemble, it is crucial to diversify the adversarial vulnerabilities among its members. This can be approached by reducing the adversarial transferability among them. To this end, we propose incorporating gradient similarity and Lipschitz regularizers into the training process. Moreover, to overcome the obstacle of a large number of parameters, we develop a parameter-lean version of RED (PS-RED). Extensive experiments, conducted across popular datasets, demonstrate that the proposed methods not only significantly improve ensemble robustness but also minimize inference delays and optimize storage usage for ensemble models. For example, our models enhance robust accuracy by approximately 15% (RED) and save parameters by approximately 90% (PS-RED) on CIFAR-10 compared with the most recent baselines.

## 1 INTRODUCTION

In recent decades, deep neural networks (DNNs) have achieved mightily impressive success in various fields: computer vision Wang et al. (2023); Xu et al. (2023), natural language processing Dao et al. (2022), speech recognition Tüske et al. (2021), graphs Wang et al. (2022). Nevertheless, we need to consider more about the robustness and stability than the precision when deploying these DNNs to real-world applications, or we will pay a heavy price for unsafe application deployment. Particularly, it is evident that almost all DNNs are susceptible to *adversarial examples* Szegedy et al. (2014), i.e., samples that can be adversarially perturbed to mislead DNNs but are very close to the original examples to be imperceptible to the human visual system. Typically, attackers generate adversarial examples by relying on gradient information to maximize the loss within a small perturbation neighborhood, which is usually referred to as the adversary's perturbation model.

How to defend against such adversarial examples has attracted remarkable attention from deep learning researchers. Lots of heuristic defenses have been proposed, e.g., adversarial training Madry et al. (2018), activation pruning Dhillon et al. (2018), and loss modification Pang et al. (2020). Besides, some scholars improved the robustness of DNNs with provable methods Wong & Kolter (2018); Cohen et al. (2019); Zhang et al. (2023). Among these defence methods, ensemble defences Tramèr et al. (2018); Kariyappa & Qureshi (2019); Pang et al. (2019); Yang et al. (2020; 2021) serve as a time-tested and effective branch of defence methods. Ensemble defences adhere to an assumption: given a set of dissimilar sub-models, the attack capabilities of adversarial examples generated by sub-model A will be weakened when using these examples to attack sub-model B, because sub-model A and sub-model B are as dissimilar as possible and the adversarial transferability between them are very low. Ensemble defences train several sub-models to jointly resist the adversarial attacks, where the sub-models need to be as dissimilar as possible to improve the ensemble robust-

ness performance. In the inference stage, tested data are fed to each sub-model and the outputs are aggregated to derive the final prediction.

In the above procedure, three challenges that hinder the ensemble defences in application of real-world systems/devices: *I*. most existing ensemble defences only achieve suboptimal robustness compared with single robust methods; *II*. the inference of ensemble defences require aggregating all the output results of all sub-models, which is latency-intensive and unsuitable for real-time devices; *III*. ensemble defences obtain a set of sub-models, which largely increases the number of parameters. In some devices, especially in some mobile devices or Internet of Things devices, such a large storage requirement is unaffordable. Thus, we are inspired to rethink the ensemble framework to boost the above aspects of performance: robustness, inference latency and parameter number. One may want to generate adversarial samples by sub-model A and use another dissimilar sub-model B to defend against them. Normally, we cannot control which sub-model the attackers use to generate adversarial examples. However, we can use the randomness strategy to confuse the attackers for improving the robustness performance of ensemble defences. Inspired by the idea of randomness, we propose a novel ensemble defence approach, termed Random Ensemble Defence (RED), to enhance the ensemble robustness (Challenge *I*) while simultaneously accelerating the inference process (Challenge *II*) by randomly sampling one member from the model ensemble for inference. In this way, each adversarial sample generated in the previous round is fed to a different sub-model with a high probability. In order to train an effective model ensemble, the members are supposed to be as diversified as possible, which can be turned into the reduction of adversarial transferability among members. Firstly, we present the gradient similarity regularizer to diversify the adversarial vulnerabilities between two sub-models. Besides, we leverage the Lipschitz continuity to derive the Lipschitz regularizer to reduce the adversarial susceptibility of every member. To address Challenge *III*, we employ the concept of hypernetwork to reduce the number of parameters and propose the Parameter-Saving version of RED (PS-RED). To be concrete, we construct a meta hypernetwork backbone for DNNs and train specialized hypernetworks to generate the parameter weights of each sub-model in the ensemble. Last but not least, we extensively demonstrate the ensemble robustness superiority and the inference as well as parameter efficiency of our proposed method by evaluating it on the latest attack methods and comparing it with existing state-of-the-art ensemble robust methods on popular benchmark datasets (CIFAR-10 and TinyImageNet). The experimental results show that our proposed method achieves substantially superior performance over all the counterparts and generalize to diverse perturbations well and significantly reduce the parameter number. Before ending this section, we summarize the contributions of this paper as follows:

- We propose a new ensemble defence method, Random Ensemble Defence (RED), to boost the ensemble robustness and speed up the inference process.
- To effectively train RED, we derive two effective regularizer: gradient similarity regularizer and Lipschitz regularizer through theoretical analysis of ensemble defences.
- We leverage the idea of hypernetworks to reduce the number of parameters and propose the parameter-saving random ensemble defence approach.
- We evaluate our proposed methods on various popular benchmarks against diverse adversarial attacks, achieving state-of-the-art performance compared with the latest counterparts.

## 2 RELATED WORK

### 2.1 ADVERSARIAL ROBUSTNESS

Numerous methods were proposed to defend against adversarial perturbation, such as Dhillon et al. (2018); Madry et al. (2018); Carmon et al. (2019); Yu et al. (2022); Lin et al. (2024), among which *adversarial training* Madry et al. (2018) and its variants Zhang et al. (2019); Wang & Zhang (2019); Stutz et al. (2020) are one of most popular and most effective methods, which aim to train a surrogate model with adversarial examples generated by projected gradient descent (PGD) Madry et al. (2018) with some norm, and then uses this surrogate model to defend against adversarial attacks. They remain quite popular since they continue to perform well in various empirical benchmarks, though it comes with no formal guarantees. More recently, some variants have been proposed to further improve the robustness performance, like input transform Guo et al. (2017); Xie et al. (2018); Li et al. (2021), revising loss functions Wang et al. (2020); Sriramanan et al. (2020); Pang et al. (2020),

adversarial data augmentation Wang et al. (2021); Rebuffi et al. (2021), provable defenses Cohen et al. (2019); Lecuyer et al. (2019); Zhang et al. (2023), adversarial weight perturbation Wu et al. (2020). Besides, Lipschitz continuity is a good consideration direction about the generalization and robustness of DNNs, and some related works Usama & Chang (2018); Khromov & Singh (2024); Chen et al. (2024) aimed at reducing the adversarial vulnerability (mainly single model defences).

## 2.2 ENSEMBLE DEFENCES

Among adversarial robustness, ensemble defences is an intriguing research direction due to their time-tested and effective robust performance. The essential point to construct an effective ensemble robust method is to reduce the adversarial transferability among the sub-models in the ensemble set. To enhance ensemble robustness, researchers presented a plenty of effective methods. For instance, Kariyappa and Qureshi Kariyappa & Qureshi (2019) minimized the cosine similarity of the gradients of sub-models to reduce adversarial transferability and improve ensemble robustness; Pang *et al.* Pang et al. (2019) used a class entropy based adaptive diversity promoting method to boost the ensemble robustness; Yang *et al.* Yang et al. (2020) presented a robust ensemble training method that diversifies the non-robust features of sub-models via an adversarial training objective function. Yang *et al.* Yang et al. (2021) offered a theoretical guarantee of adversarial transferability and empirically proposed the corresponding ensemble defence method. The presented tight empirical upper bound encourage us to establish an effective sub-model ensemble robust set.

## 2.3 HYPERNETWORKS

Proposed by Ha *et al.* Ha et al. (2017), hypernetworks are aimed at using a small network to generate the weights for a larger network (denoted as a main network). The performance the the generated network is usually the same as that of the main network with direct optimization: hypernetworks also learn to map some raw data to their desired targets, while hypernetworks take a set of embeddings that contain information about the structure of the weights and generates the weight for that layer. These switch-like pocket networks show superiority in some computer vision tasks: Oswald *et al.* von Oswald et al. (2020) utilized hypernetworks to alleviate the catastrophic forgetting phenomenon in the continual learning task; Alaluf *et al.* Alaluf et al. (2022) and Dinh *et al.* Dinh et al. (2022) simultaneously used hypernetworks to improve image editing; Peng *et al.* Peng et al. (2022) applied hypernetworks to the task of 3D medical image segmentation. In this paper, we extend the hypernetwork framework to compress ensemble robust models.

## 3 METHODOLOGY

### 3.1 RANDOM ENSEMBLE DEFENCE MODELING

In general, aggregating several individual sub-models is usually effective to enhance the performance of DNNs Russakovsky et al. (2015). Specifically, we denote the $i$-th sub-model in the ensemble set as $f_\theta^i \in \mathbb{R}^L$, representing the output of the DNN model $i$ with parameters $\theta$ over $L$ categories. Thus, the construction of the final output of the ensemble set $F$ is the direct average of all sub-models as

$$F(x) = \frac{1}{N} \sum_{i=1}^{N} f_\theta^i(x), \tag{1}$$

where $N$ is the total number of sub-models in the ensemble set. Simultaneous training is usually applied to construct the ensemble model set. Leveraging the cross-entropy loss to train the ensemble set, we can obtain the robust ensemble that can help us defend against adversarial attacks. However, if we use Eqn. (1) as the inference strategy, the final output aggregates all the outputs of all members, which reduces the adversarial robustness into the average level of the ensemble. In this way, we cannot select the most suitable sub-model to defend against a specific attack. Besides, to aggregate the outputs of all sub-models and average them is time-consuming, which is unaffordable for the extreme real-time devices and systems. To alleviate the above problems, we propose the random sampling inference (RSI) strategy, i.e., in the inference stage, only one member is sampled from the ensemble set $F$ for generating the final prediction, which can be formulated as

$$F(x) = f_\theta^{\text{Randint}(1,N)}(x), \tag{2}$$

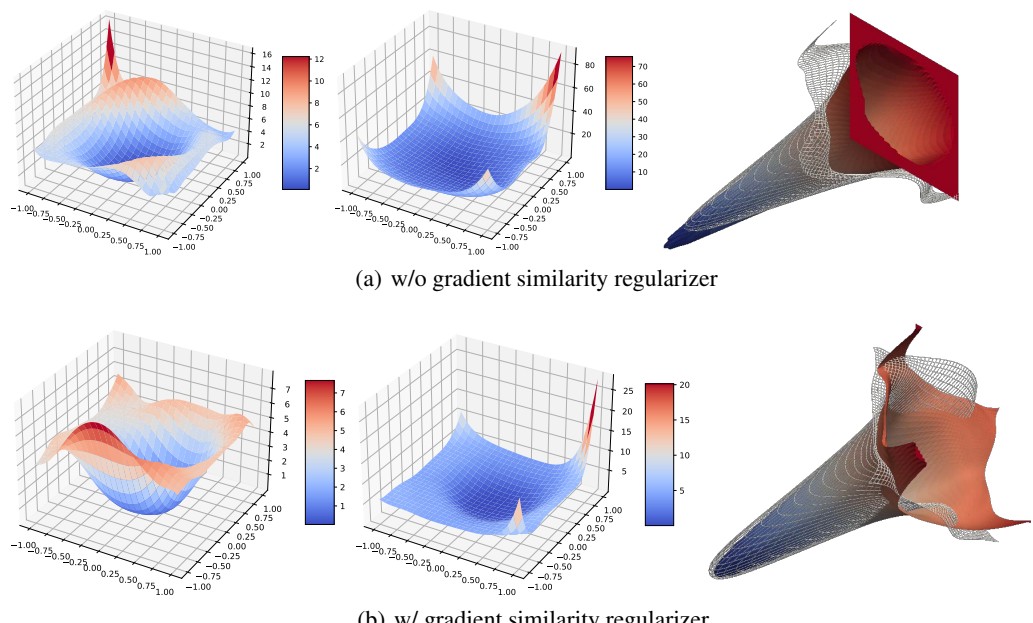

(a) w/o gradient similarity regularizer

(b) w/ gradient similarity regularizer

Figure 1: Loss landscapes of models without or with the gradient similarity regularizer. Note that the left and middle ones are the loss landscapes of two sub-models; the right one is the corresponding 3D surface of both models.

where $\mathrm{Randint}(1, N)$ is the random function that sample one integer from $1$ to $N$. The RSI strategy avoids the forward-propagation computation of the other $N - 1$ sub-models and the average operation of the $N$ sub-outputs, which speeds up the inference process by a great margin. In addition, the RSI strategy is beneficial to the improving the adversarial robustness of the robust ensemble. Because the attackers use the last output of the ensemble as the victim model to generate the adversarial examples, and then feed them to the next state of the ensemble with the RSI strategy. There is a probability of $(N - 1)/N$ that the sub-model sampled next time is different from the previous selected sub-model. However, the success of RSI depends on the consistency and similarity of the sub-models. For example, if the $N$ sub-models are the same or highly similar, RSI will fail for improving the ensemble robustness. Therefore, we should let the members in the ensemble be "disimilar". In other words, we should reduce the adversarial transferability among sub-models.

## 3.2 GRADIENT SIMILARITY AND LIPSCHITZ REGULARIZERS

Because the adversarial perturbations are normally generated with the input gradients, a direct approach to make gradients of every sub-models as dissimilar as possible. Here, we apply the widely-used cosine similarity to characterize the degree of similarity by defining a related gradient similarity regularizer as

$$\mathcal{R}_{\mathrm{sim}} = \frac{1}{C(N, 2)} \sum_{i=1}^{N} \sum_{j=1, j \neq i}^{N} \frac{|\nabla_x \ell_{f_\theta^i}(x, y) \cdot \nabla_x \ell_{f_\theta^j}(x, y)|}{\max(\|\nabla_x \ell_{f_\theta^i}(x, y)\| \cdot \|\nabla_x \ell_{f_\theta^j}(x, y)\|, \delta)}, \tag{3}$$

where $|\cdot|$ denotes the absolute operation; $\|\cdot\|$ denotes the 2-norm operation; $C(N, 2)$ is the number of combinations of $N$ taken 2 at a time; $\delta$ is a sufficiently small and positive number to prevent the denominator in the equation from equaling to zero. We visualize the loss landscapes of the two-sub-model ensemble with and without this regularizer in Fig. 1, which showcases that the gradient similarity regularizer helps reduce the entanglement of the loss surface. For example, without the regularizer, the loss surfaces are interpenetrated in the cone bottom (cf., the upper right subfigure). Besides, it is also evident that this regularizer additionally reduces the loss magnitude by a great margin, which indirectly reduces the gradient magnitude and the gradient curvature. Note that the clean data $(x, y)$ in the above equation can be replaced by the adversarial data $(\hat{x}, y)$, which further

reduces the adversarial transferability at the cost of reduced accuracy for clean data:

$$\mathcal{R}_{\text{sim}} =$$

$$\frac{1}{2C(N,2)} \sum_{i=1}^{N} \sum_{j=1, j \neq i}^{N} \left( \frac{|\nabla_x \ell_{f_\theta^i}(x,y) \cdot \nabla_x \ell_{f_\theta^j}(x,y)|}{\max(\|\nabla_x \ell_{f_\theta^i}(x,y)\| \cdot \|\nabla_x \ell_{f_\theta^j}(x,y)\|, \delta)} + \frac{|\nabla_{\hat{x}} \ell_{f_\theta^i}(\hat{x},y) \cdot \nabla_{\hat{x}} \ell_{f_\theta^j}(\hat{x},y)|}{\max(\|\nabla_{\hat{x}} \ell_{f_\theta^i}(\hat{x},y)\| \cdot \|\nabla_{\hat{x}} \ell_{f_\theta^j}(\hat{x},y)\|, \delta)} \right). \quad (4)$$

Eqn. (4) alleviates the phenomenon that adversarial examples generated by a sub-model transfer to another sub-model. Now we consider how to reduce the vulnerability of a single sub-model. Adversarial training serves as an effective and direct method to improve the robustness of single sub-model. However, in this work, we consider this question from the perspective of Lipschitz continuity. Lipschitz continuity and its application in bolstering the robustness and generalization of DNNs Usama & Chang (2018); Nguyen & Khanh (2021); Khromov & Singh (2024); Chen et al. (2024) are impressive, which are defined as: Let $\ell \colon \mathbb{R}^m \mapsto \mathbb{R}^n$ be a function defined on the metric space $\mathbb{R}^m$. The function $\ell$ is said to be Lipschitz continuous if there exists a non-negative real number $L$ (Lipschitz constant), such that for all $x_1, x_2 \in \mathbb{R}^m$, the following inequality holds

$$\|\ell(x_1) - \ell(x_2)\| \leq L\|x_1 - x_2\|, \quad (5)$$

where $\|\cdot\|$ denotes the Euclidean norm (or any other type of norm, here we use the 2-norm), and the inequality states that the change in the function's output is bounded by $L$ times the change in its input. If the function satisfies this condition, it is said to be Lipschitz continuous, and $L$ is the smallest constant for which the inequality holds. In this work, we have no intention of solving the upper and lower bounds of this NP-hard problem like Khromov & Singh (2024). Instead, we want to apply it to improving the robustness of the ensemble. Let $x_1 = x$ and $x_2 = \hat{x} = x + \Delta x$ (assume $\Delta x \neq 0$), we rewrite (5) as

$$\frac{\|\ell(x_1) - \ell(x_2)\|}{\|x_1 - x_2\|} = \frac{\|\ell(x) - \ell(\hat{x})\|}{\|x - \hat{x}\|} = \frac{\|\ell(x) - \ell(x + \Delta x)\|}{\|\Delta x\|} \leq L. \quad (6)$$

Take the limit of $\Delta x$ and leveraging the definition of derivative, we obtain:

$$\lim_{\Delta x \to 0} \frac{\|\ell(x) - \ell(x + \Delta x)\|}{\|\Delta x\|} = \|\nabla_x \ell(x)\| \leq L, \quad (7)$$

where $\nabla_x \ell(x)$ denotes the derivative (gradient) of $\ell$ w.r.t $x$. In the setting of adversarial robustness, $\ell$, $x$, $\Delta x$ and $\hat{x}$ represent some loss function of DNN models, the input, the adversarial perturbations and the adversarial examples, respectively. We can use (7) to approximate the gradient of $\ell(x)$, such that $x \gg \Delta x$. Thus, the optimization with Lipschitz constraint can be formulated as

$$\min_\theta \frac{1}{N} \sum_{i=1}^{N} \ell_{f_\theta^i}(x,y),$$
$$\text{s.b. } \|\nabla_x \ell_{f_\theta^i}(x,y)\| \leq L_i, \ i = 1, 2, \cdots, N, \quad (8)$$

where $\ell$ is some loss function (e.g., cross-entropy loss). Eqn. (8) is the non-convex optimization under complex constraints. Here, we apply the idea of Lagrangian Relaxation Gaudioso (2020) to approximately solve it:

$$\min_\theta \frac{1}{N} \sum_{i=1}^{N} \ell_{f_\theta^i}(x,y) + \sum_{i=1}^{N} \lambda_i (\|\nabla_x \ell_{f_\theta^i}(x,y)\| - L_i)$$
$$\iff \min_\theta \frac{1}{N} \sum_{i=1}^{N} \ell_{f_\theta^i}(x,y) + \sum_{i=1}^{N} \lambda_i \|\nabla_x \ell_{f_\theta^i}(x,y)\| - \sum_{i=1}^{N} \lambda_i L_i. \quad (9)$$

where $\lambda_i \geq 0$. We further approximate all $\lambda_i$ as the same value $\lambda_a/N$ (i.e., $\lambda_1 = \lambda_2 = \cdots = \lambda_N = \lambda_a/N$). Thus, Eqn. (9) is rewritten as

$$\min_\theta \frac{1}{N} \sum_{i=1}^{N} \ell_{f_\theta^i}(x,y) + \lambda_a \frac{1}{N} \sum_{i=1}^{N} \|\nabla_x \ell_{f_\theta^i}(x,y)\| - \lambda_a \frac{1}{N} \sum_{i=1}^{N} L_i. \quad (10)$$

According to Eqn. (5), $L_i$ is a (Lipschitz) constant. Thus, the item $\lambda_a \frac{1}{N} \sum_{i=1}^{N} L_i$ is also a constant, which has no influence on the optimization. Here, we simplify (10) as

$$\min_\theta \frac{1}{N} \sum_{i=1}^{N} \ell_{f_\theta^i}(x,y) + \lambda_a \frac{1}{N} \sum_{i=1}^{N} \|\nabla_x \ell_{f_\theta^i}(x,y)\|. \quad (11)$$

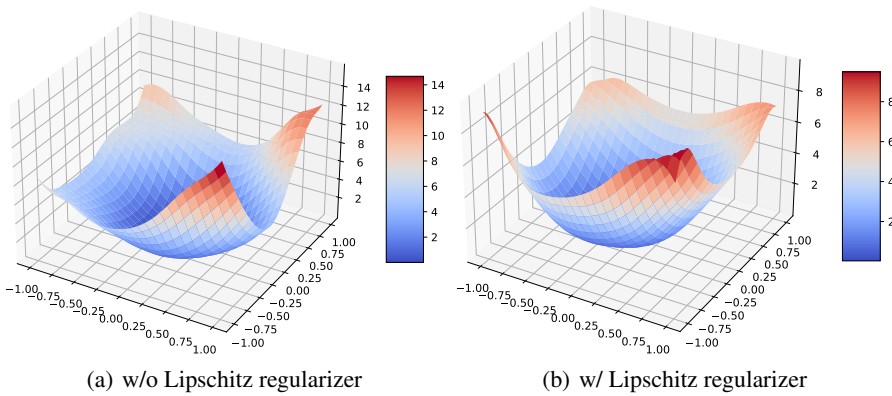

(a) w/o Lipschitz regularizer          (b) w/ Lipschitz regularizer

Figure 2: Loss landscapes of models without or with the Lipschitz regularizer.

where we refer to $\frac{1}{N}\sum_{i=1}^{N}\|\nabla_x\ell_{f_\theta^i}(x,y)\|$ as the Lipschitz regularizer. Note that Eqn. (11) helps us circumvent the solving of $L_i$ (NP-hard problem). We replace clean input by adversarial input $(\hat{x}, y)$ to obtain the full version of Lipschitz regularizer as

$$\mathcal{R}_{\text{Lipschitz}} = \frac{1}{2N}\sum_{i=1}^{N}\left(\|\nabla_x\ell_{f_\theta^i}(x,y)\| + \|\nabla_{\hat{x}}\ell_{f_\theta^i}(\hat{x},y)\|\right). \tag{12}$$

With the Lipschitz regularizer, the ensemble's members are collectively refined to achieve a higher degree of smoothness, which helps reduce the adversarial vulnerability of each sub-model as well as the adversarial transferability among sub-models in the ensemble. We conducted an observational experiment to show the performance of the Lipschitz regularizer (12), as shown in Fig. 2. The loss landscape with the Lipschitz regularizer (approximately ranging from 0 to 10) indicates the smaller loss magnitude than that without it (approximately ranging from 0 to 15). Incorporating the gradient similarity regularizer (4), the total optimization loss can be formulated as

$$\mathcal{L}_{\text{train}} = \frac{1}{N}\sum_{i=1}^{N}\ell_{f_\theta^i}^{\text{ce}}(x,y) + \lambda_a\mathcal{R}_{\text{Lipschitz}} + \lambda_b\mathcal{R}_{\text{sim}}, \tag{13}$$

where $\lambda_a$ and $\lambda_b$ are the scaling coefficients. Here, we set $\ell$ as the cross-entropy loss ($\ell^{\text{ce}}$). At this point, we have established a training strategy of a novel effective and efficient ensemble robust modeling, dubbed as Random Ensemble Defence (RED). Note that the two proposed regularizers is consistent with the theorems in Yang et al. (2021). We display the loss landscapes with the two proposed regularizers and other counterparts' loss landscapes in Appendix D.

### 3.3 PARAMETER-SAVING RED

Another disadvantage of existing ensemble defenses is that the requirement of storage for model parameters is $N$ times larger than that for single-model defenses, which is sometimes unaffordable for those compact intelligent devices. To efficiently overcome this challenge, we propose the parameter-saving random ensemble defence (PS-RED) method via hypernetworks Ha et al. (2017), which are designed to leverage a smaller network to dynamically generate weights for a larger network (denoted as the main network). This approach allows us efficiently and adaptively configure the main network's parameters, enhancing its flexibility.

### 3.3.1 OVERVIEW OF PS-RED

Specifically, instead of directly optimizing the parameters $\theta$ of a standard DNN model $f_\theta$, we train a hypernetwork $\mathcal{H}$ to generate parameters of each layer of the target classifier $f_\theta$, which can be formulated as

$$\theta_l = \mathcal{H}(z_l), \tag{14}$$

where $l$ is the $l$-layer of the DNN $f_\theta$; $z_l$ denotes the input embedding of the hypernetwork $\mathcal{H}$ for generating parameters of the $l$-layer. Here, hypernetwork is sometimes referred to weight generator.

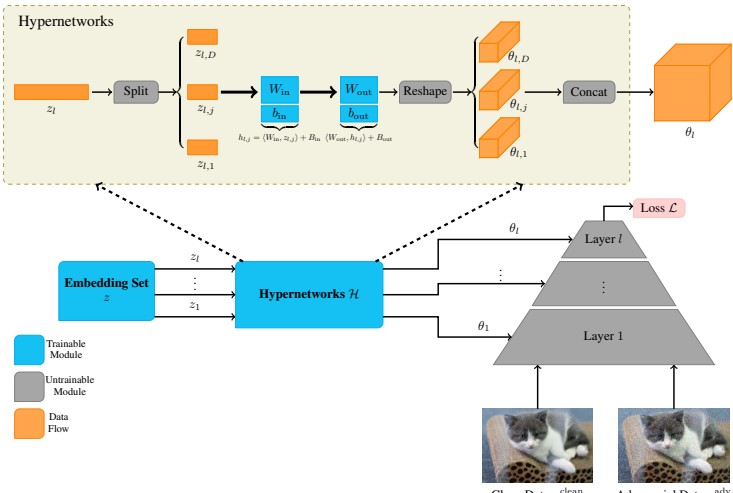

Figure 3: Overview of parameter generation via hypernetworks. Here, we leverage the embedding set and the hypernetworks to generate parameters for the main network. Note that the main network is only a forward propagation network without any optimization, while we optimize the embedding set and the hypernetworks.

Thus, the hypernetwork version of the ensemble method (1) and RED (2) can be rewritten as

$$F(x) = \frac{1}{N} \sum_{i=1}^{N} f_{\mathcal{H}^i(z^i)}(x), \tag{15}$$

$$F(x) = f_{\mathcal{H}^{\text{Randint}(1,N)}(z^i)}(x), \tag{16}$$

where $z^i = \{z^i_l\}_{l=1,\cdots,L}$ denotes the embedding set of the DNN model $i$ with $L$ layers; $\mathcal{H}^i$ represents the hypernetworks of the DNN model $i$. Because the hypernetwork version of RED (16) can save much storage required for the parameters of the ensemble method, we term it as the parameter-saving RED (PS-RED).

### 3.3.2 Hypernetwork Design

The main networks usually contain more than ten million parameters. On the one hand, controlling too many parameters would account for an infeasible model that requires huge training resources. On the other hand, every layer contains different types of parameters. Thus, we need to design a unified approach to generate parameters for every layer of the main networks, or the parameter number of the hypernetwork will increase significantly. Considering these two points, designing an expressive network is very challenging, which needs an elaborate balance between expressive performance and the selection as well as the unification of generated parameters.

**Parameter selection.** Generally, for a standard convolutional neural networks-based DNN model, there are five primary types of layers: convolution layer, activation layer, batch normalization (BN) layer, pooling layer, and fully connected layer. The activation layer and pooling layer do not contain any parameters. The widely used BN layer includes two parameters, so it is unnecessary to utilize a complicated network to generate only two parameters. Besides, the fully connected layer usually contains thousands/hundreds of times as many parameters as the number of total classes, which only accounts for a very small percentage of the total parameters. Thus, we will not generate the parameters of the fully connected layer. The last type of layer, the convolution layer, contains almost all of the parameters in a model, which is the focus of our discussion. Usually, the parameter structure of some convolution layer $l$ is $C^{\text{out}}_l \times C^{\text{in}}_l \times k \times k$, where $k \times k$ is the size of convolution kernel; $C^{\text{out}}_l$ represents the total number of filters, each with $C^{\text{in}}_l$ channels.

**Generated parameter unification.** For popular networks (like ResNet), researchers prefer $3 \times 3$ convolution kernel and choose $64K (K = 1, 2, \cdots)$ channels/filters. Thus, we design the output unit of our hypernetwork as $64 \times 64 \times 3 \times 3$ for unifying the structure of generated parameters. For parameters with larger structures, we concatenate the output unit in the filter $C^{\text{out}}_l$ and channel $C^{\text{in}}_l$

Table 1: Robust experimental results (%) of our proposed methods (RED and PS-RED) as well as four state-of-the-art counterparts under different adversarial attacks on the CIFAR-10 and TinyImageNet benchmark datasets. The best results are highlighted in **BOLD**, and the second-best results are underlined.

| Dataset | Method | NAT | FGSM | MIM | BIM | PGD | CW | DeepFool | AutoAttack |
|---------|--------|-----|------|-----|-----|-----|-----|----------|------------|
| CIFAR-10 | GAL | 95.59 | 57.63 | 8.71 | 6.06 | 5.47 | 95.25 | 18.45 | 6.29 |
| | ADP | **95.82** | 55.43 | 26.22 | 22.10 | 20.20 | **95.73** | 4.39 | 3.66 |
| | DVERGE | 92.85 | **76.52** | 37.02 | 36.05 | 34.06 | 92.77 | 38.57 | 50.97 |
| | TRS | 91.01 | 54.82 | 31.30 | 28.07 | 27.60 | 90.87 | 6.12 | 20.61 |
| | RED | 87.81 | 63.31 | **53.53** | **52.26** | **51.07** | 88.13 | **54.81** | **65.32** |
| | PS-RED | 84.01 | 56.59 | 46.63 | 44.31 | 43.21 | 79.53 | 49.43 | 53.39 |
| TinyImageNet | GAL | 66.15 | 8.64 | 1.08 | 0.86 | 1.07 | 25.95 | 13.43 | 0.06 |
| | ADP | **66.81** | 16.89 | 7.96 | 6.57 | 6.92 | 14.67 | 6.55 | 1.67 |
| | DVERGE | 63.53 | 39.53 | 19.43 | 19.05 | 17.89 | **34.24** | **43.27** | 13.31 |
| | TRS | 59.69 | 34.28 | 24.66 | 23.60 | 22.48 | 17.15 | 30.63 | 19.50 |
| | RED | 57.55 | **41.11** | **28.94** | **27.62** | **24.96** | 33.47 | 41.60 | **42.28** |
| | PS-RED | 54.74 | 38.94 | 25.88 | 24.63 | 22.32 | 26.64 | 33.72 | 37.81 |

dimensions. For some simplified parameter structures, we downsample the kernel dimensions, e.g., use the average/maximum/sum operator to downsample $3 \times 3$ kernel into $1 \times 1$ kernel. Note that we do not generate the first convolution layer $64 \times 3 \times 3 \times 3$, since the parameter number is very few compared with the total parameter number and it is not easy to generate them by the output unit.

**Hypernetwork structure.** Motivated by Ha et al. (2017), we design a two-layer linear network as our hypernetwork. The first layer takes the embedding $z_l$ as input and linearly projects it into the hidden layer. The second layer is a linear operation that takes the output of the hidden layer as input and linearly projects it into the output unit. Thus, the process that generates $l$-layer's parameters with hypernetwork can be written as

$$
\begin{aligned}
h_{l,j} &= \langle W_{\text{in}}, z_{l,j} \rangle + B_{\text{in}}, \\
\theta_{l,j} &= \langle W_{\text{out}}, h_{l,j} \rangle + B_{\text{out}}, \\
\theta_l &= \text{concat}(\theta_{l,j})_{j=1,\cdots,D}
\end{aligned}
\tag{17}
$$

where $D$ is the number of the output unit in the layer $l$; $z_l = \{z_{l,j}\}_{j=1,\cdots,D}$ denotes the $l$-layer embedding with $D$ sub-embeddings. Here the trainable parameters of hypernetwork $\mathcal{H}$ are $W_{\text{in}}$, $B_{\text{in}}$, $W_{\text{out}}$ and $B_{\text{out}}$. Additionally, the layer embeddings $z = \{z_l\}_{l=1,\cdots,L}$ ($L$ is the number of total generated convolution layers) are also learnable. Note that parameters of different layers in the main networks are generated by a shared hypernetwork and corresponding embeddings. We summarize the parameter generation process in Fig. 3.

## 4 EXPERIMENTS

### 4.1 EXPERIMENTAL SETUP

**Datasets.** We conducted extensive experiments on the below two datasets: CIFAR-10 Krizhevsky (2009) and TinyImageNet Russakovsky et al. (2015). CIFAR-10 includes $50,000$ in training images and $10,000$ in test images with 10 classes. TinyImageNet is the subset of ImageNet Russakovsky et al. (2015) dataset, containing 500 training images, 50 validation images, and 50 test images for each class (the total number of classes is 200), respectively. Images on CIFAR-10 are sized $32 \times 32$, and images on TinyImageNet are with a size of $64 \times 64$.

**Implementation details and baselines.** We utilized ResNet-18 He et al. (2016) architecture as a basic network for CIFAR-10 and TinyImageNet datasets due to computational resource limit. Besides, we selected 128 as the dimension of the hypernetwork embeddings. For most experiments, we selected the model number $N$ as 8 for CIFAR-10 and 3 for TinyImageNet; for the ablation

Table 2: Further robust experimental results (%) of our proposed methods (RED and PS-RED) as well as the counterparts under different black-box and white-box adversarial attacks on CIFAR-10. The best results are highlighted in **BOLD**, and the second-best results are underlined.

| Method | OnePixel | Square | Pixle | DI2-FGSM | EoT-PGD | APGD | SparseFool |
|--------|----------|--------|-------|----------|---------|------|-----------|
| GAL | 84.13 | 69.15 | 1.18 | 12.10 | 5.69 | 7.41 | 15.40 |
| ADP | **86.51** | 70.80 | 0.76 | 18.97 | 9.15 | 5.34 | 14.12 |
| DVERGE | 86.36 | 83.23 | 5.49 | 34.24 | 31.24 | 31.50 | 17.52 |
| TRS | 85.75 | 74.49 | 4.69 | 30.08 | 27.63 | 24.06 | 28.38 |
| RED | 83.44 | **85.16** | **68.79** | **45.13** | **44.13** | **62.19** | **36.05** |
| PS-RED | 77.15 | 77.32 | 54.23 | 37.22 | 37.89 | 48.77 | 30.73 |

experiments for the model number, we let $N$ be 3, 5, 8 as well as 12, respectively. We compared our proposed method with state-of-the-art ensemble methods: GAL Kariyappa & Qureshi (2019), ADP Pang et al. (2019), DVERGE Yang et al. (2020) and TRS Yang et al. (2021). Note that we implement the baseline methods with the hyper-parameters claimed in their original papers. For our methods, we set the scaling hyper-parameters $\lambda_a$ and $\lambda_b$ as 10 and 10, respectively. Due to the space limit, we put more experimental setup details in Appendix A.

## 4.2 MAIN ROBUSTNESS RESULTS

In this subsection, we present the main white-box robustness evaluation results of our proposed ensemble methods and the counterparts. Due to the space limit, extensive ablation study experimental results are showed in Appendix B.

**Results on CIFAR-10.** The robust experimental results on the small dataset (CIFAR-10) are shown in Tab. 1. As observed from the table, it is evident that our proposed ensemble methods outperform the baselines under diverse adversarial settings. For example, RED achieves better adversarial accuracy for the seen adversarial attacks: RED obtains more than $15\%$, $15\%$ and $16\%$ increments under the classical gradient-based attacks, MIM, BIM and PGD, respectively. We primarily attribute this superiority to the negative interference of our proposed RSI strategy on the acquisition of adversarial perturbations. Besides, our proposed RED achieves more than $11\%$ and $25\%$ gains under the strong unseen DeepFool and AutoAttack attacks, respectively. This means that our RED method plays a beneficial role in defending against unseen attacks. Although the performance of the proposed parameter-efficient PS-RED is somewhat inferior to RED's performance, PS-RED also has a commendable performance on defending various adversarial attacks. For instance, among all the ensemble methods, PS-RED achieves the second-best performance when defending against the MIM, BIM, PGD, DeepFool and AutoAttack attacks. The reason its robust performance is not as strong as RED is primarily due to the hypernetworks' tendency to weaken the neural network's representation power, which occurs when they uniformly generate a set of weights with reduced flexibility. Moreover, PS-RED still outperforms the baselines under most adversarial settings, which mainly attributes to the RSI mechanism that increases the cost for attackers to generate adversarial samples. Note that for RED and PS-RED, the AutoAttack accuracy is higher than the PGD accuracy, while others do the opposite. It is partly because the AutoAttack overfits the current sub-model for RED and PS-RED, if the next inference sample another member, the attack performance of generated perturbations is greatly weakened. Another possible explanation is that the AutoAttack stops once the effective adversarial examples are generated against the current sub-model (they may not be effective for other sub-models), while the PGD only stops when preset iterations are reached. Thus, the adversarial examples by PGD are stronger than those by AutoAttack.

**Results on TinyImageNet.** We also provide more experiments on the TinyImageNet dataset to evaluate the performance of our proposed methods on large datasets. The results are shown in Tab. 1, from which it is evident that our proposed methods achieve better adversarial robustness than that of the baseline methods under most adversarial settings, namely, RED accomplishes the best or second-best robust performance for all listed attacks. For instance, RED achieved the best adversarial accuracy for the FGSM evaluation, while DVERGE is the most effective ensemble defence on the CIFAR-10 benchmark; for the MIM, BIM and PGD evaluations, RED increases about $4\%$

Table 3: Robust experimental results (%) of our proposed RED and PS-RED plus adversarial training. The best results of every method are stressed in **BOLD**.

| Method | NAT | FGSM | MIM | PGD | AutoAttack |
|---|---|---|---|---|---|
| RED | **87.81** | 63.31 | 53.53 | 51.07 | 65.32 |
| +adv. training | 77.99 | **68.21** | **65.06** | **61.12** | **71.62** |
| PS-RED | **84.01** | 56.59 | 46.63 | 43.21 | 53.39 |
| +adv. training | 68.84 | **62.83** | **48.57** | **46.37** | **56.13** |

of adversarial accuracy compared with the baselines. For the AutoAttack evaluation, RED further improve by more than 22 percentage points. Besides, PS-RED also achieves notable robustness, especially for MIM, BIM and AutoAttack.

### 4.3 FURTHER ANALYSIS

**Further evaluation.** We further employ several black-box attacks (OnePixel, Pixle, Square and DI2-FGSM) and white-box attacks (EoT-PGD and SparseFool) to evaluate the adversarial performance of our proposed RED and PS-RED methods as well as the baselines. The hyper-parameter configurations for all these attacks adhere to the default settings specified in the TorchAttacks package and are therefore not detailed here. The experimental results are displayed in Tab. 2. For the black-box OnePixel attack evaluation, our proposed methods do not show the superiority compared with the baselines. However, for all other further evaluations, our proposed methods achieved better robustness, to a greater or lesser degree. For example, RED achieves more than $5\%$, $11\%$, and $12\%$ increments that outperform the best robustness of the baselines for the Square, DI2-FGSM and EoT-PGD evaluations, respectively. Particularly, RED achieves more than $60\%$ increment for the Pixle evaluation. Besides, PS-RED also achieves about $50\%$ increment for the Pixle evaluation. This means than the baselines lack robustness against the Pixle attack, while our PSI strategy has a high degree of robustness against the Pixle attack. Furthermore, in the evaluation of SparseFool, our novel RED and PS-RED methods secure the first and second positions in terms of robustness performance, respectively.

**Plus adversarial training.** We also conduct experiments that combine our proposed ensemble methods with the notable adversarial training method. The results are shown in Tab. 3. From the table, it is evident that adversarial training greatly boost our ensemble methods, especially for the MIM and PGD evaluation. For example, the adversarial accuracy of RED increases more than 12% and 10% for the MIM and PGD evaluations, respectively; the adversarial accuracy of PS-RED increases more than 2% and 3% for the MIM and PGD evaluations, respectively. For AutoAttack, RED with adversarial training achieves about 6% increment, and PS-RED with adversarial training achieves about 3% increment, respectively. For more analytical evaluations, please refer to Appendix C.

## 5 CONCLUSION

In this paper, to boost the ensemble robustness with simultaneously accelerating the inference process, we turned the effectiveness problem of ensemble defences into the reduction problem of adversarial transferability among members in the ensemble, and further introduced an innovative random sampling inference strategy with two effective training regularizers (gradient similarity and Lipschitz) and proposed the corresponding random ensemble defence (RED) method. Additionally, by leveraging the idea of hypernetworks, we further proposed the parameter-saving version of RED (PS-RED) for reducing the storage requirement of ensemble models. Last but no least, we conducted comprehensive experiments to validate the superiority of the proposed RED as well as PS-RED methods under diverse strong white-box and black-box attacks, which achieve better robust results compared to existing state-of-the-art ensemble defence methods across widely adopted benchmark datasets.

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
