## A MORE EXPERIMENTAL SETUP DETAILS

**Training parameters.** For training stages, we used SGD optimizer with learning rate 0.1, momentum 0.9 and weight decay $5 \times 10^{-4}$ to train our models and all baselines with the step decay learning rate strategy. We trained all the models with 120 epochs with one NVIDIA GeForce RTX 4090 GPU, and the learning rate will decrease to 0.01 and 0.001 at epochs 60 and 110, respectively. We set the batch size as 256 for CIFAR-10 and 128 for TinyImageNet, respectively. For adversarial setting (PGD) on CIFAR-10, we set the maximum perturbations $\epsilon$ as 0.03, the step sizes $\alpha$ as 0.0075 and the attack step $T$ as 10. Besides, on TinyImageNet, we set $\epsilon$ as 0.015, $\alpha$ as 0.00375 and $T$ as 4.

**Evaluation.** We employ a variety of adversarial attacks to assess the performance of our proposed methods as well as compare our methods with state-of-the-art baselines: clean evaluation (NAT), white-box attacks, like Fast Gradient Sign Method (FGSM) Goodfellow et al. (2015), Momentum Iterative Method (MIM) Dong et al. (2018), Basic Iterative Method (BIM) Kurakin et al. (2018), Projected Gradient Descent (PGD) Madry et al. (2018), Carlini & Wanger Attack (C&W) Carlini & Wagner (2017), DeepFool Attack (DeepFool) Moosavi-Dezfooli et al. (2016), AutoAttack (AutoAttack) Croce & Hein (2020), Expectation over Transformation PGD (EoT-PGD) Athalye et al. (2018), adaptive PGD (APGD) Croce & Hein (2020) and SparseFool Attack (SparseFool) Modas et al. (2019); blakc-box attacks, such as OnePixel Attack (OnePixel) Su et al. (2019), Square Attack (Square) Andriushchenko et al. (2020), Pixel Attack (Pixel) Pomponi et al. (2022) and DI2-FGSM Attack (DI2-FGSM) Xie et al. (2019). For those that require setting the parameter of the attack strength, we uniformly set the attack strength to 0.02 for the CIFAR-10 dataset as well as 0.01 for the TinyImageNet dataset. Furthermore, we set the attack step size as 0.005 for CIFAR-10 and 0.0025 for TinyImageNet. The attack iterations of MIM, BIN and PGD are set to be 10. Due to the space limit, for a large number of other parameters of all the attacks, we follow the default setting in the TorchAttacks package Kim (2020), a PyTorch library that provides adversarial attacks to generate adversarial examples.

## B ABLATION STUDY EXPERIMENTAL RESULTS

In this appendix section, we show the results about comprehensive ablation studies: inference method, the functions of two regularizer losses, the embedding dimension of the learnable embedding input for hypernetworks, and the number of sub-models in the ensemble.

Table 4: Inference method ablation experimental results of our proposed RED and PS-RED on CIFAR-10. "AVG" means the traditional average inference strategy. Better results for every training method are highlighted in **BOLD**.

| Dataset | Method | Inference | NAT | FGSM | MIM | PGD | AutoAttack |
|---------|--------|-----------|-----|------|-----|-----|------------|
| CIFAR-10 | RED | AVG | **91.52** | 57.05 | 32.40 | 28.97 | 22.63 |
| | | RSI | 87.81 | **63.31** | **53.53** | **51.07** | **65.32** |
| | PS-RED | AVG | **84.29** | 47.20 | 28.72 | 26.67 | 18.94 |
| | | RSI | 84.01 | **56.59** | **46.63** | **43.21** | **53.39** |
| TinyImageNet | RED | AVG | **60.40** | 32.86 | 23.50 | 21.89 | 19.29 |
| | | RSI | 57.55 | **41.11** | **28.94** | **24.96** | **42.28** |
| | PS-RED | AVG | **57.98** | 36.68 | 18.75 | 16.35 | 14.57 |
| | | RSI | 54.74 | **38.94** | **25.88** | **22.32** | **37.81** |

### B.1 INFERENCE METHOD ABLATION EXPERIMENTS

We provide the ablation experimental results to evaluate the function of the RSI strategy in Tab. 4. As seen in the table, we can see that the RSI strategy greatly improve the ensemble robustness. Especially for AutoAttack, the RED/PS-RED improve the adversarial accuracy by ∼40% compared

with the traditional average strategy. We attribute this increment to that the RSI strategy interrupts the coherence of generating powerful adversarial samples.

Table 5: Lipschitz regularizer ablation study results (%) of our proposed ensemble methods on CIFAR10. The best results are stressed in **BOLD**.

| Method | $\lambda_a$ | $\lambda_b$ | NAT | FGSM | MIM | PGD | AutoAttack |
|---|---|---|---|---|---|---|---|
| RED | 0 | 10 | **90.69** | 39.27 | 16.05 | 18.57 | 29.75 |
|  | 1 | 10 | 88.76 | 49.79 | 26.08 | 31.12 | 48.73 |
|  | 10 | 10 | 87.81 | **63.31** | **53.53** | **51.07** | 65.32 |
|  | 100 | 10 | 77.20 | 59.74 | 39.49 | 42.05 | **66.83** |
| PS-RED | 0 | 10 | **90.47** | 24.08 | 1.58 | 1.93 | 8.41 |
|  | 1 | 10 | 85.76 | 39.42 | 9.87 | 13.15 | 34.66 |
|  | 10 | 10 | 84.01 | **56.59** | **46.63** | **43.21** | **53.39** |
|  | 100 | 10 | 66.52 | 32.34 | 19.06 | 29.68 | 41.09 |

## B.2 LIPSCHITZ REGULARIZER ABLATION EXPERIMENTS

For evaluating the function of the Lipschitz regularizer $\mathcal{R}_{\text{Lipschitz}}$, we conduct the Lipschitz regularizer ablation study with letting $\lambda_b = 10$ (for $\mathcal{R}_{\text{sim}}$). The experimental results are displayed in Tab. 5, form which it is manifest that appropriate settings of $\lambda_a$ (for $\mathcal{R}_{\text{Lipschitz}}$) assist us reach better robust performance of our proposed RED and PS-RED. Too small value of $\lambda_a$ boosts the adversarial accuracy for the NAT evaluation, e.g., when $\lambda_a = 0$, the RED method achieves the best results (90.69%) under clear input. Besides, excessively large value of $\lambda_a$ does not bring additional gains, for example, when $\lambda_a = 100$, the robustness is inferior to that with $\lambda_a = 10$ against FGSM, MIM and PGD.

Table 6: Gradient similarity regularizer ablation study results (%) of our proposed ensemble methods on CIFAR10. The best results are stressed in **BOLD**.

| Method | $\lambda_a$ | $\lambda_b$ | NAT | FGSM | MIM | PGD | AutoAttack |
|---|---|---|---|---|---|---|---|
| RED | 10 | 0 | 79.95 | 54.13 | 38.19 | 31.19 | 41.19 |
|  | 10 | 1 | 82.97 | 60.75 | 42.98 | 38.77 | 54.29 |
|  | 10 | 10 | **87.81** | **63.31** | **53.53** | **51.07** | **65.32** |
|  | 10 | 100 | 71.93 | 57.98 | 40.51 | 46.90 | 63.73 |
| PS-RED | 10 | 0 | 82.74 | 38.39 | 33.43 | 30.46 | 36.02 |
|  | 10 | 1 | **88.35** | 47.23 | 40.80 | 33.08 | 39.14 |
|  | 10 | 10 | 84.01 | **56.59** | **46.63** | **43.21** | **53.39** |
|  | 10 | 100 | 69.30 | 49.32 | 33.25 | 35.98 | 52.00 |

## B.3 GRADIENT SIMILARITY REGULARIZER ABLATION EXPERIMENTS

To assess the function of the gradient similarity regularizer $\mathcal{R}_{\text{sim}}$, we perform the gradient similarity regularizer ablation study with letting $\lambda_a = 10$ (for $\mathcal{R}_{\text{Lipschitz}}$), whose results are showcased in Tab. 6. According to the table, it is evident that proper assignments of $\lambda_b$ (for $\mathcal{R}_{\text{sim}}$) help us achieve better robust performance of our proposed RED and PS-RED. Too small $\lambda_b$ significantly weakens the performance of the proposed methods, for example, when $\lambda_b = 0$, the robustness results of RED

and PS-RED both decrease by a great margin. Nevertheless, too large value of $\lambda_b$ provides few gain for the ensemble robustness. Thus, we set $\lambda_b$ as 10 for better robustness performance of our proposed methods.

## B.4 MEMBER NUMBER ABLATION EXPERIMENTS

To evaluate the influence of the member number in the ensemble set, we provide the robustness results of our proposed methods and baselines with different ensemble members (3, 5, 8, 12). The results are shown in Tab. 7, from which we can safely conclude that for most ensemble methods, more members in the ensemble implies better adversarial robustness; for a specific number of members, our proposed RED and PS-RED demonstrate stronger ensemble robustness compared with the baselines. There is a exception, i.e., TRS, whose eight-member version is inferior to its five-member version.

Table 7: Member number ablation study results (%) of our proposed methods and baselines on CIFAR-10. The best results of every method are stressed in **BOLD**.

| Method | # of Member | NAT | FGSM | MIM | PGD | AutoAttack |
|---|---|---|---|---|---|---|
| GAL | 3 | 94.24 | 18.02 | 1.42 | 0.78 | 0.00 |
| | 5 | 94.68 | 48.43 | 6.59 | 5.72 | 1.79 |
| | 8 | **95.59** | 57.63 | 8.71 | 5.47 | 6.29 |
| | 12 | 95.58 | **58.18** | **9.49** | **5.91** | **7.18** |
| APD | 3 | 94.17 | 57.59 | 2.26 | 0.81 | 2.63 |
| | 5 | 94.82 | 60.75 | 17.19 | 9.15 | 2.23 |
| | 8 | **95.82** | 55.43 | 26.22 | 20.20 | 3.66 |
| | 12 | 94.99 | **57.92** | **26.77** | **23.11** | **5.22** |
| DVERGE | 3 | 92.83 | 68.55 | 18.51 | 12.85 | 20.62 |
| | 5 | 93.28 | **75.73** | 30.02 | 26.38 | 42.14 |
| | 8 | 92.85 | 76.52 | 37.02 | 34.06 | 50.97 |
| | 12 | **93.04** | 69.88 | **46.56** | **44.40** | **53.25** |
| TRS | 3 | 86.73 | 44.03 | 18.63 | 14.22 | 2.48 |
| | 5 | 88.08 | 55.57 | 34.49 | 31.31 | **24.41** |
| | 8 | **91.01** | 54.82 | 31.30 | 27.60 | 20.61 |
| | 12 | 89.12 | **55.68** | **34.95** | **32.69** | 22.50 |
| RED | 3 | 86.76 | 57.15 | 29.08 | 26.78 | 48.91 |
| | 5 | 87.09 | 57.62 | 45.46 | 42.01 | 59.15 |
| | 8 | 87.81 | 63.31 | 53.53 | 51.07 | 65.32 |
| | 12 | **88.28** | **65.30** | **54.62** | **53.69** | **67.57** |
| PS-RED | 3 | 82.62 | 52.82 | 22.64 | 16.18 | 39.31 |
| | 5 | 83.09 | 53.01 | 38.76 | 30.81 | 44.50 |
| | 8 | 84.01 | 56.59 | 46.63 | 43.21 | 53.39 |
| | 12 | **85.93** | **57.41** | **48.44** | **46.60** | **55.85** |

## B.5 EMBEDDING DIMENSION ABLATION EXPERIMENTS

The original ensemble models (i.e., GAL, ADP, DVERGE, TRS and RED) contain 89,392,208 and 33,837,336 parameters for 8-sub-model CIFAR-10 ensemble and 3-sub-model TinyImageNet en-

semble, respectively; while the parameter-efficient PS-RED methods (128 embedding dimension) respectively include 9,528,400 and 3,865,560 parameters, saving approximately 90% of the parameters. To explore the effect of parameter savings and robustness performance of the PS-RED method, we conduct an ablation study on the hypernetworks input embedding. The experimental results are depicted in Tab. 8. Note that we compare the number of parameters between the original ensemble models (i.e., GAL, ADP, DVERGE, TRS and RED) and the PS-RED model with a specific embedding dimension (like 32, 64, 128 and 256). As listed in the table, there exists a trade-off between the embedding dimension and the robustness performance: normally, larger embedding dimension provides better robustness, which is probably because larger embedding dimension of the input embeddings can enhance the representational capacity of hypernetworks, thus improving the adversarial robustness. Nonetheless, a higher embedding dimension implies a need for more storage space in PS-RED models. To illustrate, the number of parameters with the embedding dimension of 256 is more than 39 times as that with the embedding dimension of 32. In addition, too large dimension does not generate too many extra profits. As a example, the parameter count with the embedding dimension of 256 is approximately three times greater than that with the embedding dimension of 128, while the robustness only improves about 3% and 2% for the MIM and PGD evaluations, respectively. Thus, the selection of hypernetworks input embedding dimension requires a multifaceted trade-off and consideration.

Table 8: Hypernetworks' input embedding dimension ablation study results (%) of the proposed PS-RED method on CIFAR10.

| Embed. | # of Para. | Para. Saving | NAT | FGSM | MIM | PGD | AutoAttack |
|---|---|---|---|---|---|---|---|
| 32 | 911,440 | ↓98.98% | 70.55 | 51.84 | 25.04 | 19.59 | 45.42 |
| 64 | 2,735,184 | ↓96.94% | 83.86 | 58.19 | 32.88 | 29.01 | 46.48 |
| 128 | 9,528,400 | ↓89.34% | 84.01 | 56.59 | 46.63 | 43.21 | 53.39 |
| 256 | 35,697,744 | ↓60.07% | 90.53 | 61.76 | 49.41 | 45.55 | 57.94 |

## C  MORE ANALYTICAL EXPERIMENTAL RESULTS

We conducted a more in-depth analysis of our proposed RED and PS-RED from various aspects: multiple execution and comparison with latest random ensemble methods. These further experiments assist readers gain a deeper insight into the proposed ensemble methods.

Table 9: Accuracy means and standard deviations (%) of our methods and baselines on CIFAR-10 under multiple executions. The best results are highlighted in **BOLD**, and the second-best results are underlined.

| Method | NAT | FGSM | MIM | BIM | PGD | C&W | DeepFool | AutoAttack |
|---|---|---|---|---|---|---|---|---|
| GAL | **94.64±1.16** | 57.12±0.79 | 8.83±0.25 | 6.19±0.40 | 5.27±0.31 | 93.39±4.23 | 17.24±1.49 | 6.32±0.29 |
| ADP | 93.60±2.25 | 56.21±1.22 | 25.55±0.70 | 22.42±0.93 | 19.63±0.57 | **94.35±3.82** | 4.66±0.36 | 4.09±0.70 |
| DVERGE | 92.19±1.01 | **75.42±2.12** | 37.57±0.58 | 35.71±0.69 | 33.04±1.05 | 93.04±1.18 | 39.20±0.75 | 50.45±1.02 |
| TRS | 91.25±1.23 | 55.06±0.37 | 30.88±0.53 | 28.48±1.21 | 27.11±0.71 | 90.89±2.02 | 7.71±1.62 | 20.52±0.99 |
| RED | 88.33±2.31 | 62.21±1.24 | **54.58±1.29** | **52.73±1.47** | **50.50±0.67** | 87.54±2.59 | **52.72±2.33** | **64.48±1.11** |
| PS-RED | 82.47±2.83 | 54.30±2.77 | 45.81±1.82 | 44.49±1.18 | 41.61±1.89 | 82.04±3.51 | 49.33±0.24 | 52.68±0.71 |

### C.1  MULTIPLE EXECUTION EXPERIMENTS

Our ensemble methods include a random module (RSI), whose performance may be influenced by different execution environments. To eliminate the environmental variance and get a stable robustness evaluation, we run our proposed RED as well as PS-RED multiple times (five times). The means and standard deviations are shown in Tab. 9, in which we used the different machine random seed for the multi-execution results. As indicated in the table, it is clear that our proposed methods can obtain stable robustness results that are consistently close to the robustness results in Tab. 1. Additionally, we also conduct multi-execution comparative experiments of the corresponding coun-

terparts, as illustrated in Tab. 9. The comparison results reveal that our proposed RED and PS-RED excel over the baselines with a stable and noticeable robustness increments.

## C.2 LATEST RANDOM ENSEMBLE METHOD COMPARISON

Cai *et al.* Cai et al. (2023) presented the random gated networks and ensemble-in-one (EIO) method to boost ensemble robustness, achieving state-of-the-art ensemble robustness. We compared our proposed methods with theirs. We used the default setting with their published code and train 3-path and 8-path EIO super-nets. For fair comparison, we trained 3-member and 8-member RED and PS-RED ensemble models. Results against white-box attacks are showcased in Tab. 10. The table clearly demonstrates that our approach achieves a similar clean accuracy compared to EIO. However, our method's robustness significantly surpasses EIO's, particularly for the 8-member/path model. We believe EIO's deficiency stems from inadequate training as the number of paths increases, coupled with a lack of explicit constraints to ensure diversity in adversarial transferability across paths. Within EIO's framework, only one path is trained per step, which compounds the training challenge with an increased number of paths. Furthermore, despite utilizing distilled adversarial examples to enhance the diversity of each path's vulnerability, it is challenging to sample all paths adequately to improve their collective vulnerability diversity due to the sheer number of potential paths. In addition, we have presented experimental outcomes that incorporate adversarial training. For models with 3 members or paths, the EIO framework, when enhanced with adversarial training, exhibits superior robustness in comparison to our RED and PS-RED approaches. However, as previously discussed, an increased number of paths does not necessarily translate to enhanced robustness. On the contrary, our methods with 8 members demonstrate improved robustness over both the 8-path-EIO and the 3-path-EIO models. This suggests that while adversarial training can bolster EIO's performance with a smaller number of paths, our approach is more effective in achieving robustness with a larger number of members or paths.

Table 10: Comparison Robust experimental results (%) of our proposed RED and PS-RED with the latest EIO on CIFAR-10. The best results of every setting are stressed in **BOLD**.

| Member/Path | Method | NAT | FGSM | MIM | PGD | AutoAttack |
|---|---|---|---|---|---|---|
| 3 | EIO | **87.22** | 20.29 | 7.12 | 5.81 | 3.78 |
| | RED | 86.76 | **57.15** | **29.08** | **26.78** | **48.91** |
| | PS-RED | 82.62 | 52.82 | 22.64 | 16.18 | 39.31 |
| | EIO+adv. training | **82.00** | **55.12** | **51.49** | **50.77** | 48.29 |
| | RED+adv. training | 78.53 | 51.68 | 45.98 | 44.34 | **52.12** |
| | PS-RED+adv. training | 76.01 | 48.43 | 40.20 | 38.92 | 44.78 |
| 8 | EIO | 86.52 | 18.87 | 7.40 | 6.29 | 4.41 |
| | RED | **87.81** | **63.31** | **53.53** | **51.07** | **65.32** |
| | PS-RED | 84.01 | 56.59 | 46.63 | 43.21 | 53.39 |
| | EIO+adv. training | **81.18** | 54.46 | 50.14 | 49.32 | 45.39 |
| | RED+adv. training | 77.99 | **68.21** | **65.06** | **61.12** | **71.62** |
| | PS-RED+adv. training | 68.84 | 62.83 | 48.57 | 46.37 | 56.13 |

## D LOSS LANDSCAPES OF PROPOSED AND BASELINE METHODS

In this appendix section, we used the the neural network visualization tool provided by Li et al. (2018) to visualize the loss landscapes of our proposed methods and counterparts in Figs. 4 and 5.

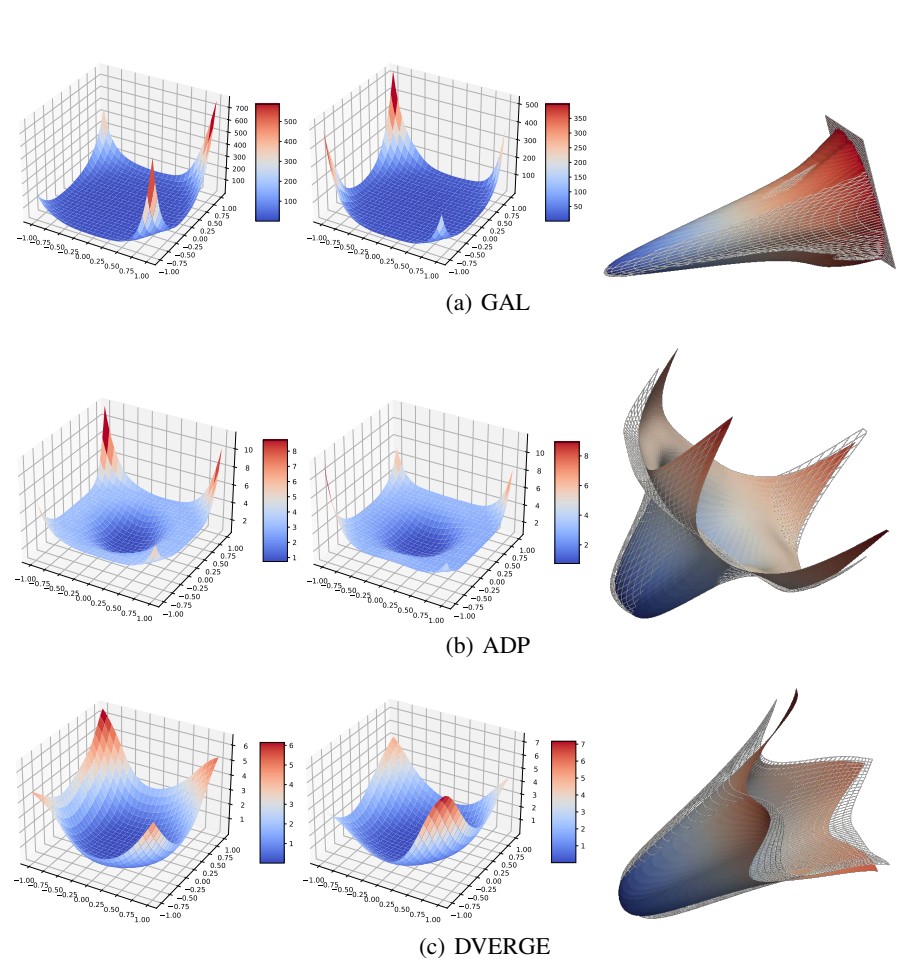

(a) GAL

(b) ADP

(c) DVERGE

Figure 4: Loss landscapes with different training methods (GAL, ADP, DVERGE). The left and middle ones are the loss landscapes of two sub-models; the right one is the corresponding 3D surface of both models.

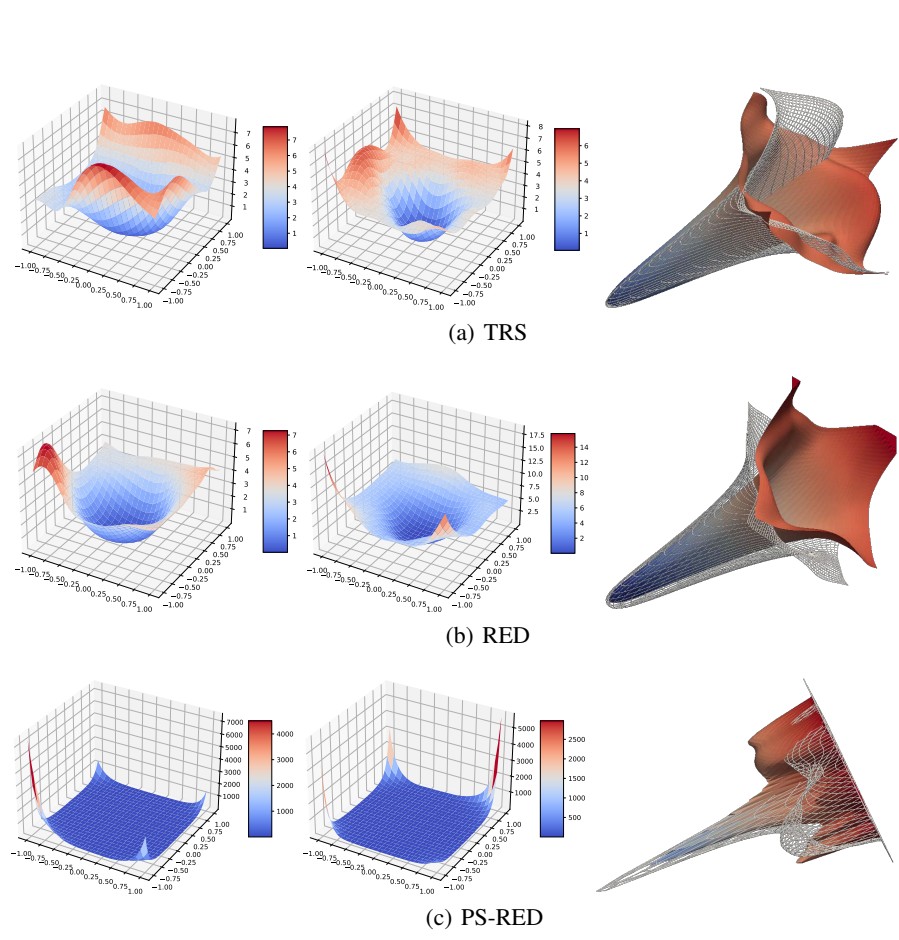

(a) TRS

(b) RED

(c) PS-RED

Figure 5: Loss landscapes with different training methods (TRS, RED, PS-RED). The left and middle ones are the loss landscapes of two sub-models; the right one is the corresponding 3D surface of both models.