# OpenReview forum: "RED: Efficiently Boosting Ensemble Robustness via Random Sampling Inference"
_ICLR.cc/2025/Conference — ICLR 2025 Conference Withdrawn Submission_

### Official Review · Reviewer_Bbj7 · 2024-10-21

**Soundness:** 3
**Presentation:** 3
**Contribution:** 2
**Rating:** 6
**Confidence:** 4

**Summary:**

This paper introduces Random Ensemble Defense (RED), an ensemble defense method to enhance the robustness of DNNs against adversarial attacks while reducing inference latency and model size. RED employs random sampling during inference to improve efficiency and make attacks harder. To further strengthen the ensemble, the authors use gradient similarity and Lipschitz regularizers to diversify vulnerabilities among models, reducing adversarial transferability. They also propose a lightweight version called PS-RED, which reduces the number of parameters. Experiments on CIFAR-10 demonstrate that RED improves robust accuracy by 15% and PS-RED reduces parameter size by 90% compared to recent baselines.

**Strengths:**

The strengths of this paper include:

- The writing is clear and uses intuitive visualization to motivate the proposed methods.
- Ensemble-based defenses are an important research topic, and the proposed RED is simple and straightforward to implement. PS-RED's design is interesting to me.
- It's reasonable to use gradient similarity to diversify vulnerabilities among models, and use Lipschitz regularizers to make individual model more robust.
- I'm surprised to see that in Table 1, a ResNet-18 model can achieve 65.32% acc under AutoAttack. If this result is correct, the effectiveness of RED is really promising.

**Weaknesses:**

The strengths of this paper include:

- The RED (randomly sampling a model) design, as well as the gradient similarity and Lipschitz regularizer terms, are all fairly simple. I haven't kept up with the latest advances, but I believe there have be many defense works that use similar ideas, making this paper's technical contributions less significant.

- According to Eq. (13), it seems that RED does not use adversarial training. I wonder if RED is compatible with advanced adversarial training methods (e.g., those listed on RobustBench); besides, if using models larger than ResNet-18, can RED achieve higher (or even SOTA) acc under AutoAttack?

**Questions:**

Please see the Weaknesses section.

---

### Official Review · Reviewer_CTGz · 2024-11-03

**Soundness:** 3
**Presentation:** 3
**Contribution:** 2
**Rating:** 5
**Confidence:** 4

**Summary:**

The paper introduces the Random Ensemble Defense (RED) method and its parameter-saving version (PS-RED) aimed at enhancing the robustness and efficiency of ensemble defenses against adversarial attacks. The authors propose a novel approach leveraging random sampling for inference and hypernetwork-based parameter reduction. While the conceptual innovation and empirical results are commendable, there are several aspects that undermine the efficacy and novelty of the proposed methods.

**Strengths:**

1. The introduction of RED to enhance ensemble robustness by utilizing random sampling during inference is innovative. This approach reduces inference latency, which is crucial for real-time applications.
2. The experiments conducted across popular benchmarks such as CIFAR-10 and TinyImageNet demonstrate significant improvements over existing methods, validating the effectiveness of the proposed RED and PS-RED methods in enhancing ensemble robustness and operational efficiency.
3. The development of PS-RED, which uses a hypernetwork to generate model parameters, reduces the storage requirements for ensemble models, addressing a critical issue in deploying deep learning models on resource-constrained devices.

**Weaknesses:**

1. The core idea of selecting subsets from an ensemble model to enhance defense capabilities has similarities with previous works on ensemble-based transfer attacks [1].
2. The effectiveness of hypernetworks proposed might be contingent on the types of network backbones used. It is essential to evaluate the hypernetworks’ performance across a variety of architectures to ascertain their general utility.
3. The manuscript lacks a thorough comparative analysis with the latest ensemble-based defense methods [2].
4. For real-time devices, it is critical to evaluate not just the parameter efficiency but also the runtime performance of the PS-RED. The manuscript should include runtime comparisons to support claims of suitability for real-time applications.

[1]  Meta-gradient adversarial attack. ICCV 2021.
[2]  Understanding and improving ensemble adversarial defense. NeurIPS 2023.

**Questions:**

See the weeknesses above.

---

### Official Review · Reviewer_kgZB · 2024-11-04

**Soundness:** 2
**Presentation:** 2
**Contribution:** 2
**Rating:** 3
**Confidence:** 4

**Summary:**

This paper introduces Random Ensemble Defense (RED), a novel method to enhance DNN robustness against adversarial attacks by leveraging random sampling to reduce inference latency and adversarial transferability among models. PS-RED is proposed to further minimize model size with improvements in robustness and storage efficiency on CIFAR-10 compared to existing methods.

**Strengths:**

- I think the proposed method could be considered as a novel contribution,
if this paper can better differentiate itself from existing methods
that employ similar ideas.
Please see the weaknesses section for more details.

- The results appear to be promising.

**Weaknesses:**

- This paper's proposed similarity-based regularization
is very similar to the ones proposed in its references,
specifically GAL.
For instance,
GAL minimizes the cosine similarity
between the gradients of sub-models,
Could you elaborate the design rationale of eq. 3?
In addition,
the underlying objective of eq. 4 appears to be similar
to DVERGE.
It is not clear
how the core contribution of the proposed method
is different from these existing methods.

- I am not convinced that random sampling inference (RSI)
is effective in improving robustness.
This appears to use randomness as a form of gradient masking.
Unfortunately,
no ablation study regarding the above weaknesses
are found in the main submission.

- It is surprising that AutoAttack is not as effective
as PGD, C&W, BIM, MIM, and even FGSM,
as AutoAttack includes a momentum-based PGD in its toolbox,
which is typically more effective than vanilla PGD.
This suggests that either AutoAttack is not properly tuned
or the proposed method uses gradient masking
to appear robust.
This paper is also missing stronger adversarial attacks
that specifically target ensemble models,
such as MORA [a].

- Regarding the loss landscape figures,
could you elaborate on the meanings of axes
and the right-most plots?

[a]: Yu et al., MORA: Improving Ensemble Robustness Evaluation with Model-Reweighing Attack. NeurIPS 2022. https://arxiv.org/abs/2211.08008

**Questions:**

- While adversarial training can improve robustness,
Is ensemble models
with adversarially-trained sub-models more robust
under an increasing number of sub-models?
[a] seems to suggest that forming larger ensembles
with adversarially-trained DVERGE models
can surprisingly degrade robustness.

- How effective is the proposed method under black-box transferability attacks?

---

### Official Review · Reviewer_pQ3F · 2024-11-07

**Soundness:** 2
**Presentation:** 3
**Contribution:** 2
**Rating:** 3
**Confidence:** 4

**Summary:**

This paper proposes an ensemble defense method named Random Ensemble Defense (RED). The objective is to improve the robustness and efficiency of ensemble deep learning models in adversarial settings. RED utilizes a random sampling inference (RSI) strategy to accelerate inference and reduce adversarial transferability between models. The paper also proposes a parameter-saving version, PS-RED, by incorporating a hypernetwork to predict model weights.

**Strengths:**

The paper aims to improve the efficiency and robustness of ensemble deep learning models. The proposed method itself is quite novel. Though the concept of hypernetwork is not new, the usage of hypernetwork to predict model weights in the ensemble to reduce the model saving cost while enabling robustness is also quite novel.

**Weaknesses:**

1. The main objective of this paper is still to enhance the robustness. In this case, what if the attacker directly attacks the hypernetwork? As the hypernetwork has an even simpler model architecture, injecting attacks might not be challenging. But this paper does not study the case when the hypernetwork itself is attacked.

2. Limitation and cost concerns of the hypernetwork. In the paper, the hypernetwork is used to predict each layer's weight. The prediction is restricted to certain model architectures and weight shapes, which limits the practicality of the method. Furthermore, training a hypernetwork to predict weights also takes additional effort. It involves the collection of the data for hypernetwork training and the actual training time of the hypernetwork. It's unclear how much the hyperparameter training will cause.

3. Limited evaluations. Current experiments are only conducted on the ResNet18 model architecture, which is mainly composed of 3x3 CONV layers that the hypernetwork predicts. Only showing the results for one model architecture type is insufficient. Also, it's unclear whether this method can work on other model architectures with multiple different layer types.

**Questions:**

1. What is the performance of the model with the predicted model weights from the hypernetwork? Can it reach high accuracy performance on clean examples? The weight prediction seems to be conducted in a layer-by-layer manner without conditions on prior layers' weights. In this case, how to guarantee the predicted model weights together reach good performance?

2. What if the attacker directly attacks the hypernetwork?

3. Can you verify the performance of the model on other model architectures such as VGG family, mobilenet, or transformer models?

4. How is the hypernetwork trained? What is the dataset and loss term for the training of the hypernetwork?

5. What is the total training cost of the hypernetwork?

---

### Official Review · Reviewer_42so · 2024-11-07

**Soundness:** 2
**Presentation:** 3
**Contribution:** 2
**Rating:** 3
**Confidence:** 3

**Summary:**

This submission proposes a defence for adversarial evasion attacks for deep neural networks named RED. The defence is built upon random sub-model sampling, diversity regularization, Lipschitz regularization, and hypernetworks (for efficiency). Experiments on CIFAR-10 and TinyImageNet demonstrate the effectiveness against a range of state-of-the-art attacks.

**Strengths:**

- An end-to-end ensemble-based defence that achieves superior performance against existing adversarial attacks, with novel application of hypernetworks for saving the training cost.

- Good writing quality and easy to follow.

**Weaknesses:**

- Principally, my biggest concern is the lack of adaptive attack evaluation:

Though the submission evaluates the effectiveness against several state-of-the-art attacks, these attacks are not adaptive in terms of not being tailored for the random model selection mechanism. Since the attacker knows that the RED framework selects a random model each time, the attacker can query multiple times and maintain a corresponding adversarial example for each model to iterate within the ensemble. In this way, though at a higher query cost, the attack strategy is boiled down to maintaining a set of PGD iterations and should be more effective than directly applying existing attacks. As a result, the evaluation result may give a false sense of robustness.

- Another concern is the lack of novelty. The ideas of Lipschitz regularization and diversity promotion have long existed. Though the submission proposes different formats, like using $\hat x$ along with $x$, the difference is relatively minor. The hypernetwork is novel to be applied here, but may not suffice at a top-tier conference like ICLR.

Minor:
1. Line 119, Line 122, etc: citation format can be improved. Maybe try `\citet{xxx}`.

2. Would be good to have some discussion on limitations and broader societal impacts.

**Questions:**

1. In Figure 1, it seems that with and without the gradient similarity regularizer, the loss landscapes do not differ that much?
2. In line 421, are you also using model number $N=8$ or $3$ for baseline methods? Some baseline methods may involve pairwise regularizers, and how do you resolve the efficiency issues when $N$ becomes as large as 8?

---

### Note · Authors · 2024-11-20

I have read and agree with the venue's withdrawal policy on behalf of myself and my co-authors.